# Investigating the Drivers of Supply Chain Resilience in the Wake of the COVID-19 Pandemic: Empirical Evidence from an Emerging Economy

Mohammad Ali Yamin 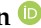

Department of Human Resources Management, Collage of Business, University of Jeddah, Jeddah 23454, Saudi Arabia; mayameen@uj.edu.sa

**Abstract:** The COVID-19 pandemic has disrupted supply chain operations globally. Nevertheless, resilient firms have the capacity to combat an unprecedented situation with the right strategic approach. The current research has developed an integrated research model that combines factors such as supply chain intelligence, supply chain communication, leadership commitment, risk management orientation, supply chain capability and network complexity to investigate supply chain resilience. The research model of this study was empirically tested with 309 responses collected from supply chain managers. Results revealed that supply chain resilience is measured with supply chain intelligence, supply chain communication, leadership commitment, risk management orientation, supply chain capability and network complexity and demonstrated a substantial variance $R^2$ of 0.548% towards supply chain resilience. Practically, this study suggests that supply chain managers should focus on factors such as big data analytics, risk management orientation,1 supply chain communication and leadership commitment to enhance supply chain resilience and sustainable supply chain performance.

**Keywords:** supply chain resilience; supply chain intelligence; communication; leadership commitment; big data analytics; sustainable supply chain performance

## 1. Introduction

In this dynamic environment, the major challenge for supply chain practitioners is to deal with supply chain upheavals, disruption, and unforeseen events [1]. If a firm faces upheavals in supply chain operations and continue to perform, that situation is characterized by resilience [2]. Supply chain resilience is defined as operational capacity of a firm to return to its initial state after being disrupted and be stronger than before in a supply chain process [3]. The importance of supply chain resilience is highlighted in earlier studies [1,3–5]. According to Brandon-Jones et al. [1] firms are facing more disruption due to natural and manmade events and therefore resilience phenomenon should be investigated to understand how resilience help firms to recover quickly after disruption. In current situation wherein COVID-19 pandemic has left devastated impact global economy and badly affected supply chain operations and therefore organizations are now seeking resilient kind of strategies to confront unforeseen events [6,7].

Concerning supply chain disruption, the literature indicates that the pandemic has destabilized supply chain operations and negatively impacted customer needs requirements and satisfaction [5,8,9]. Therefore, the current study fills a research gap and develops an integrated supply chain resilience model with a combination of factors such as supply chain intelligence, supply chain communication, leadership commitment, risk management orientation, supply chain capability, network complexity and big data analytics to investigate supply chain resilience during the COVID-19 pandemic and sustainability supply chain performance in a post-pandemic context. To enhance supply chain resilience,

the researcher has paid attention to supply chain intelligence and communication strategies. Supply chain intelligence is a process of integrating knowledge that is derived from suppliers, customers and competitors and using that knowledge to manage supply chain operations [10,11]. Therefore, communication is the extent to which a firm facilitates supply chain partners through communication, messages, and communication networks [12]. The impact of leadership commitment is found to be positive in measuring supply chain resilience. For instance, authors such as Speier, Whipple, Closs, and Voss [13] postulated that leadership commitment has been used as a foundation in designing and implementing supply chain strategies. Similarly, prior research conducted by Wieland and Marcus Wallenburg [14] confirmed that risk orientation reduces the failure chances in the supply chain process. Furthermore, supply chain capability and network complexity have shown a positive influence on measuring supply chain resilience [15,16].

The research model as shown in Figure 1 is extended with the moderating role of big data analytics. Big data analytics is identified as a combination of technologies, processes and techniques that enable organizations to collect, organize, visualize and analyze data and bring swiftness into supply chain operations [17]. The moderating effect of big data analytics is examined between supply chain resilience and sustainable supply chain performance. To the best of the authors knowledge, this study is the first that integrates factors such as leadership, technology and network factors altogether to investigate supply chain resilience and sustainable supply chain performance. This study is significant as it investigates the role of supply chain resilience and sustainable supply chain performance during the COVID-19 pandemic. In addition to that, the results of this research disclose several useful findings for the manufacturing industry to understand how to bring resilience in supply chain operations whenever they confront unforeseen events. The remaining part of the research is included in the literature review, methodology, data analysis, discussion and the conclusion of this study.

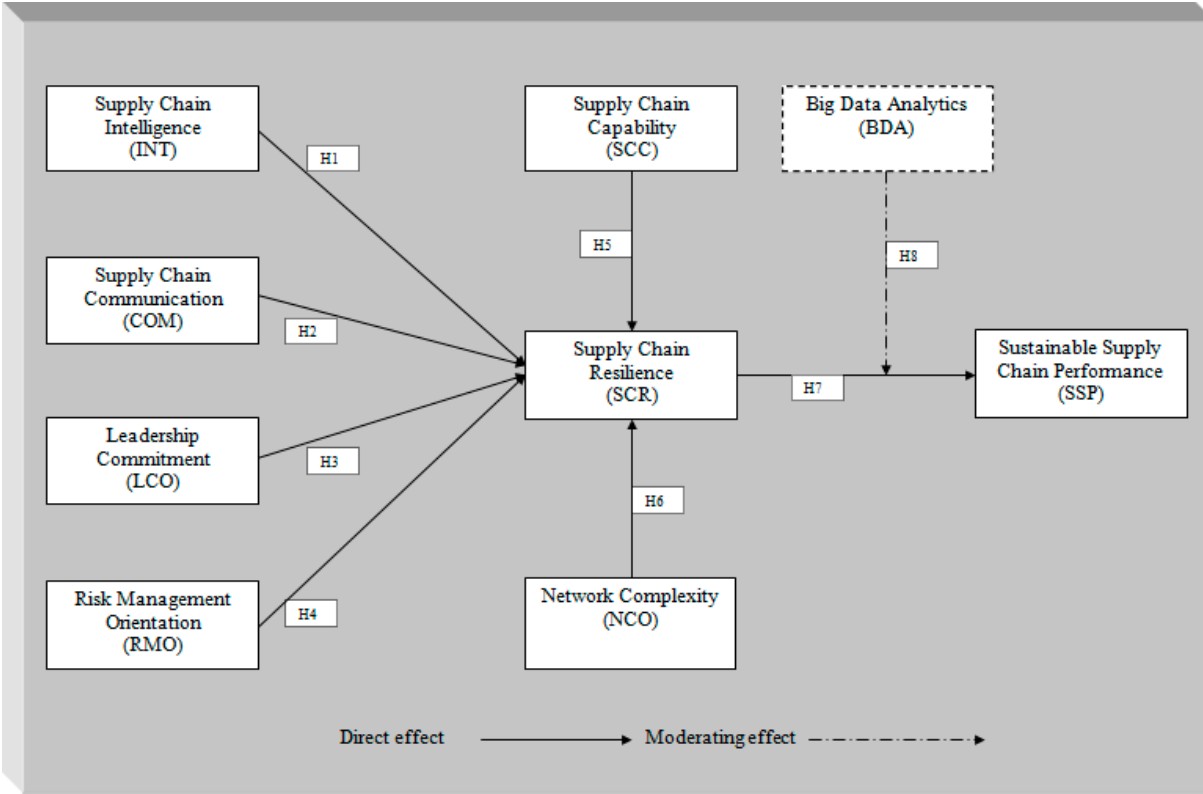

**Figure 1.** Research framework.

## 2. Literature Review

### 2.1. Supply Chain Intelligence and Communication

The concept of supply chain intelligence is extracted from knowledge-based resources and is explained as the extent to which a firm integrates knowledge that is derived from suppliers, customers and competitors and uses this to manage supply chain operations [11]. Supply chain intelligence is a process of knowledge integration among supply chain partners [11,18]. The use of a supply chain intelligence strategy gives a holistic view of the supply chain process and reduces disruption in supply chain operations [18–20]. Earlier studies have established that intelligence characteristics enable organizations to forecast accurately, reduce risk and make firms more resilient in response to supply chain uncertainty [18,19,21]. Therefore, communication is the extent to which a firm facilitates supply chain partners through communication, messages and communication networks [12] The literature has shown that intelligence and communication bring supply chain integration, responsiveness, information exchange and leverage to a higher supply chain performance [12,21,22]. A recent study conducted by Asamoah et al. [22] has confirmed that both communication and supply chain intelligence positively impact supply chain resilience. Therefore, the following hypotheses are proposed:

**Hypothesis 1 (H1).** *Supply chain intelligence significantly impacts SC resilience.*

**Hypothesis 2 (H2).** *Supply chain communication significantly impacts SC resilience.*

### 2.2. The Role of Leadership Commitment and Risk Management Orientation

The importance of leadership commitment is vital in planning, designing and implementing a supply chain strategy [15]. As suggested by Speier et al. [13] leadership commitment is considered as a foundation in the supply chain process. The literature has confirmed that a leader's commitment motivates employees, brings pro-activeness and ensures that resources are being used adequately [13,23–25]. It is established that leadership commitment assists supply chain managers in implementing decisions transparently to avoid supply chain disruption, which, in turn, enhances resilience in operations [26,27]. Concerning risk orientation, earlier studies have confirmed that risk orientation reduces the failure chances in a supply chain process [14]. Organizations can handle uncertainty in a supply chain process through risk orientation, which ultimately boosts supply chain resilience [28–30]. Therefore, the following hypotheses are proposed:

**Hypothesis 3 (H3).** *Leadership commitment significantly impacts SC resilience.*

**Hypothesis 4 (H4).** *Risk management orientation significantly impacts SC resilience.*

### 2.3. Supply Chain Capability and Network Complexity

Supply chain capability denotes to the ability of a firm to identify, manage, and synchronize information that facilitates a firms internal and external supply chain operations [16]. The literature provides abundant evidence that the supply chain capability improves a firm's operational and financial performance by reducing cost and enhancing resilience in supply chain operations [2,16,31]. Therefore, in the current research, we centered our attention toward the supply chain capability and resilience. According to Nishat Faisal, Banwet, and Shankar [32] strong coordination is required among supply chain partners to manage, forecast and replenish inventory. Authors such as Bhamra et al. [2] asserted that supply chain capability plays a vital role in the flow of goods, reduces lead time, brings transparency in supply chain operations, speeds up the payment cycle, reduces inventory and avoids over production. The network complexity is identified as the degree of connectivity between the network length and the number of nodes [15]. The complexity of the network is increased with an increase in the number of nodes and network length [4,13]. Thus, the researcher assumed that supply chain resilience may enhance with a decrease

in the numbers of nodes and the network length. Therefore, the following hypotheses are proposed:

**Hypothesis 5 (H5).** *Supply chain capability significantly impacts supply chain resilience.*

**Hypothesis 6 (H6).** *Decrease in network complexity significantly impacts supply chain resilience.*

### 2.4. Big Data Analytics

Big data analytics is an emerging concept and is being acknowledged as a process that enables organizations to collect, process, store and analyze data further to obtain useful insights [17]. In a supply chain setting, big data analytics is conceptualized as a combination of technologies, processes and techniques that enable organizations to collect, organize, visualize and analyze data and bring swiftness to their supply chain operations [17]. The extant literature has confirmed the significant influence of big data in reducing SC disruption while increasing SC resilience [33–38]. According to Janssen et al. [34] data analytics is significantly related to the supply chain innovation process. Another study conducted by Dubey et al. [17] has confirmed that big data analytics positively impacts the predication of supply chain resilience. Following the above arguments and supported by [17]. the current research tests the moderating effect of BDA on the relationship of supply chain resilience and sustainable supply chain performance. Thus, the following hypotheses are proposed:

**Hypothesis 7 (H7).** *Resilience significantly impacts sustainable SC performance.*

**Hypothesis 8 (H8).** *Big data analytics has a moderating impact on SC resilience and sustainable SC performance.*

### 3. Research Methods

#### 3.1. Designing Questionnaire and Instrument Development

In line with research objectives, the research framework of this study is developed under a quantitative and cross-sectional research approach. The researcher has followed a positivist paradigm to design the research in line with [39]. In addition to this, data are collected through research questionnaires. The research questionnaire in this study comprises construct items and demographic characteristics of the respondents. Construct items were developed by reviewing the literature and previously established scales. Scale items for network complexity were adopted from [15]. Risk management orientation scale is adopted from Wieland and Marcus Wallenburg [14] and then slightly adapted into the current research context. Supply chain resilience scale items were adopted from a previously developed scale by Brandon-Jones et al. [1] then slightly adapted. Scale items for the construct big data analytics were adopted from [17]. Supply chain communication scale items were adopted from [21]. Similarly, supply chain intelligence items were adopted from [22]. Therefore, supply chain capability scale was adapted from [16,22]. Scale items for leadership commitment were adapted from [7]. Construct items for sustainable supply chain performance were adopted from [40,41]. The scale items are measured using a 7-point Likert scale, where 1 denotes strongly disagree and 7 denotes strongly agree [42].

#### 3.2. Sampling and Data Collection

This study uses an empirically tested research framework to examine supply chain resilience and sustainable supply chain performance. Therefore, manufacturing firms are selected as the unit of analysis in this research. Earlier studies have emphasized that manufacturing companies provide in depth understanding about supply chain operations and organizational performance [2,16,31]. The population of this research comprises supply chain managers working in manufacturing companies in Saudi Arabia. As suggested by M. Yamin [43] and consistent with prior power analysis, a sample of 300 respondents was

selected for data analysis. Research data of this study were collected through an online research survey. Data were collected during the month of May 2021. The research survey was designed and conducted during COVID-19 pandemic; consequently, the researcher used online tools such as social media platforms, Google forms and email for data collection. Concerning respondent selection, the researcher selected respondents through a convenience sampling approach. For data collection, 750 questionnaires were forwarded to supply chain managers with a request to fill out an online research questionnaire according to their knowledge about supply chain resilience and sustainable supply chain performance. In response to the online research survey, 317 questionnaires were retrieved from respondents. Nonetheless, 8 questionnaires were discarded due to inadequate and inappropriate filling. Thus, a total of 309 responses were used for inferential analysis. Descriptive analyses were conducted with SPSS software. The results of the descriptive analysis revealed that the data set comprised 198 males and 111 females among 309 respondents. Similarly, the respondents' ages were considered. The results indicate that the majority of the respondents 143 were found to be between 21 and 30 years. Next to this, 35 respondents were found to be between 41 and 50 years old. Furthermore, 131 respondents were aged between 31 and 40 years. Finally, data were estimated with a structural equation modeling (SEM) approach.

## 4. Data Analysis

### 4.1. Common Method Bias

The common method variance bias (CMV) needed to be examined before the inferential analysis to ensure that the research data had no bias [44]. Since the data were collected from a single source, a CMV issue may occur in this research [45,46]. Referring to the CMV issue, the literature has suggested two well-known methods, namely, procedural and statistical remedies [45,47–49]. Following the procedural method, the questionnaire items were jumbled up, consistent with [48]. Therefore, CMV is statistically confirmed with Harman's single factor solution. Harman's single factor solution suggests that the variance explained by the first factor must not be higher than 50% [48]. The results, as depicted in Table 1, indicate that the variance explained by the first factor is less than the threshold value (50%), hence confirming that CMV is not a potential issue in this study.

**Table 1.** Harman's single factor analysis.

| | Total Variance Explained | | | | | |
|---|---|---|---|---|---|---|
| Factors | Eigenvalues | | | Extraction Sums of Squared Loadings | | |
| | Total | % of Variance | Cumulative % | Total | % of Variance | Cumulative % |
| 1 | 9.422 | 23.737 | 23.737 | 9.422 | 23.737 | 23.737 |

### 4.2. Structural Equation Modeling

The theoretical framework of this study was tested with a structural equation modeling (SEM) approach [42]. The researcher opted for a two-stage approach for the structural equation modeling computation, which included the assessment of a measurement model and a structural model [43,50]. The reliability and validity of the constructs was tested with a measurement model [43]. Therefore, the hypothesized relationship was confirmed with a structural model [51]. The data were evaluated with Smart PLS software using a partial least square (PLS) approach [52].

#### 4.2.1. Assessing Measurement Model

The measurement model was examined to establish the construct's validity and reliability. The convergent validity of the constructs was achieved with average variance extracted (AVE) following the criterion that that AVE values must be higher than 0.50. Nevertheless, the construct's reliability was achieved following the values of ($\alpha$) and composite reliability in line with Mohammad Ali Yousef Yamin and Sweiss [53] who

suggested that the CR and Cronbach's alpha values should be higher than 0.70, reflecting adequate construct reliability and validity. The findings, as depicted in Table 2, revealed the adequate reliability, validity and convergent validity of the constructs.

**Table 2.** Measurement model.

| Indicator | Loadings | (α) | CR | AVE |
|---|---|---|---|---|
| **Big Data Analytics (BDA)** | | | | |
| BDA1: In this firm, advance data analytics tools are used to take decisions. | 0.824 | 0.757 | 0.861 | 0.673 |
| BDA2: In this firm, information is extracted using big data analytics to take decision. | 0.843 | | | |
| BDA3: For data visualization, this firm use dashboard display to assist supply chain managers. | 0.793 | | | |
| **Supply Chain Communication (COM)** | | | | |
| COM1: This firm has multiple communication channels to facilitate supply chain operations. | 0.826 | 0.854 | 0.902 | 0.696 |
| COM2: This firm uses an integrated organizational system to communicate with stakeholders. | 0.841 | | | |
| COM3: This firm uses the latest integrated communication tools for channel communication. | 0.825 | | | |
| COM4: The use of frequent communication among supply chain partners enhances firm resilience. | 0.845 | | | |
| **Supply Chain Intelligence (INT)** | | | | |
| INT1: This firm is able to search, retrieve and store business information to boost supply chain operation. | 0.717 | 0.700 | 0.834 | 0.628 |
| INT3: This firm is ability to understand sales trends and customer preferences using integrated supply chain tools. | 0.869 | | | |
| INT4: This firm uses integrated information retrieved from past events to deal with any kind of unprecedented situation. | 0.784 | | | |
| **Leadership Commitment (LCO)** | | | | |
| LCO1: The leadership of this organization is committed to handling all kinds of profit and loss. | 0.752 | 0.785 | 0.860 | 0.606 |
| LCO2: Leaders of this organization take responsibility for all the departments to tackle with unprecedented situation. | 0.798 | | | |
| LCO3: Leaders of this organization support long term organizational goals. | 0.752 | | | |
| LCO4: Leadership of this organization shows pro-activeness to recover business operations. | 0.809 | | | |
| **Network Complexity (NCO)** | | | | |
| NCO2: This organization invests heavily on infrastructure to reduce network complexity. | 0.802 | 0.788 | 0.876 | 0.702 |
| NCO3: In this organization, network complexity occurred due to unexpected changes in supply chain operations. | 0.853 | | | |
| NCO4: This organization has a strategic plan to deal with supply chain nodes that reduce network complexity. | 0.858 | | | |
| **Risk Management Orientation (RMO)** | | | | |
| RMO2: Risks in this organization are monitored continuously and managed proactively. | 0.748 | 0.736 | 0.849 | 0.652 |
| RMO3: This organization has the ability to identify the source of disruption in a systematic way. | 0.820 | | | |
| RMO4: This organization is efficient in assessing own risk, customer risk and supplier risk. | 0.852 | | | |
| **Supply Chain Capability (SCC)** | | | | |
| SCC1: In this firm, the information flow is more effective between the firm and supply chain partners. | 0.796 | 0.848 | 0.898 | 0.687 |
| SCC2: This firm has the capacity to handle follow-up activities proactively. | 0.836 | | | |
| SCC3: This firm has strong coordination with stake holders for planning and forecasting. | 0.850 | | | |
| SCC4: This firm has the competency to respond quickly to changing customer needs and demands. | 0.832 | | | |
| **Supply Chain Resilience (SCR)** | | | | |
| SCR1: This firm has the competency to recover supply chain operations quickly. | 0.845 | 0.876 | 0.915 | 0.729 |
| SCR2: In this firm, inventory flow would not take long to restore. | 0.863 | | | |
| SCR3: This firm is able to restore operating performance. | 0.867 | | | |
| SCR4: This firm has the capacity to deal with all kinds of supply chain disruption without any delay. | 0.840 | | | |
| **Sustainable Supply Chain Performance (SSP)** | | | | |
| SSP1: This firm has reduced buffer stock in the supply chain process. | 0.882 | 0.851 | 0.910 | 0.770 |
| SSP2: This firm is following all environmental standards according to customer requirements. | 0.889 | | | |
| SSP3: This firm has controlled the supply chain wastage significantly. | 0.862 | | | |

The measurement model had established the construct's convergent validity and reliability. Therefore, Fornell and Larcker analysis was incorporated for the assessment of the construct's discriminant validity [24,54]. The Fornell and Larcker analysis suggests that the square root values of the average variance extracted must be higher than the other constructs correlations [24]. The findings indicate that the square root of AVE is higher

when compared with other constructs correlations, thus establishing the discriminant validity of the measure. The findings of the Fornell and Larcker analysis are tabulated in Table 3.

**Table 3.** Discriminant validity.

|  | BDA | COM | INT | LCO | NCO | RMO | SCC | SCR | SSP |
|---|---|---|---|---|---|---|---|---|---|
| BDA | 0.820 | | | | | | | | |
| COM | 0.299 | 0.834 | | | | | | | |
| INT | 0.465 | 0.244 | 0.792 | | | | | | |
| LCO | 0.400 | 0.323 | 0.386 | 0.778 | | | | | |
| NCO | 0.595 | 0.277 | 0.513 | 0.396 | 0.838 | | | | |
| RMO | 0.390 | 0.315 | 0.384 | 0.897 | 0.352 | 0.808 | | | |
| SCC | 0.271 | 0.706 | 0.262 | 0.318 | 0.279 | 0.325 | 0.829 | | |
| SCR | 0.407 | 0.470 | 0.408 | 0.653 | 0.367 | 0.662 | 0.455 | 0.854 | |
| SSP | 0.398 | 0.500 | 0.346 | 0.543 | 0.335 | 0.519 | 0.471 | 0.719 | 0.878 |

Although the Fornell and Larcker analysis was used extensively, it has some deficiencies in computation [54–56]. The cross-loading method was used in this study as an alternative method and consistent with earlier studies by [54]. The cross-loading method suggests that the indicator loading of the construct must be higher than corresponding constructs loading [57]. The results of the cross-loading method revealed that construct loadings are satisfactory when compared with the corresponding constructs loading and, therefore, establish the discriminant validity of the constructs. The results of the cross loading are exhibited in Table 4.

**Table 4.** Cross loadings.

|  | BDA | COM | INT | LCO | NCO | RMO | SCC | SCR | SSP |
|---|---|---|---|---|---|---|---|---|---|
| BDA1 | 0.824 | 0.232 | 0.388 | 0.360 | 0.564 | 0.371 | 0.257 | 0.365 | 0.352 |
| BDA2 | 0.843 | 0.273 | 0.332 | 0.298 | 0.442 | 0.284 | 0.229 | 0.306 | 0.311 |
| BDA3 | 0.793 | 0.233 | 0.422 | 0.322 | 0.450 | 0.298 | 0.177 | 0.325 | 0.312 |
| COM1 | 0.212 | 0.826 | 0.200 | 0.320 | 0.218 | 0.288 | 0.562 | 0.391 | 0.432 |
| COM2 | 0.242 | 0.841 | 0.186 | 0.264 | 0.195 | 0.266 | 0.533 | 0.415 | 0.409 |
| COM3 | 0.295 | 0.825 | 0.221 | 0.293 | 0.254 | 0.298 | 0.619 | 0.377 | 0.427 |
| COM4 | 0.252 | 0.845 | 0.210 | 0.202 | 0.259 | 0.199 | 0.646 | 0.384 | 0.400 |
| INT1 | 0.342 | 0.187 | 0.717 | 0.268 | 0.327 | 0.263 | 0.233 | 0.287 | 0.270 |
| INT3 | 0.409 | 0.190 | 0.869 | 0.328 | 0.433 | 0.332 | 0.187 | 0.352 | 0.282 |
| INT4 | 0.351 | 0.205 | 0.784 | 0.317 | 0.451 | 0.313 | 0.210 | 0.327 | 0.274 |
| LCO1 | 0.309 | 0.258 | 0.296 | 0.752 | 0.361 | 0.540 | 0.245 | 0.433 | 0.385 |
| LCO2 | 0.345 | 0.262 | 0.329 | 0.798 | 0.340 | 0.700 | 0.233 | 0.464 | 0.415 |
| LCO3 | 0.256 | 0.208 | 0.239 | 0.752 | 0.219 | 0.725 | 0.186 | 0.497 | 0.415 |
| LCO4 | 0.335 | 0.276 | 0.333 | 0.809 | 0.323 | 0.793 | 0.312 | 0.608 | 0.464 |
| NCO2 | 0.391 | 0.228 | 0.439 | 0.292 | 0.802 | 0.287 | 0.245 | 0.288 | 0.299 |
| NCO3 | 0.491 | 0.195 | 0.445 | 0.316 | 0.853 | 0.274 | 0.214 | 0.290 | 0.244 |
| NCO4 | 0.599 | 0.267 | 0.410 | 0.381 | 0.858 | 0.321 | 0.242 | 0.340 | 0.297 |
| RMO2 | 0.310 | 0.257 | 0.324 | 0.711 | 0.291 | 0.748 | 0.237 | 0.429 | 0.388 |
| RMO3 | 0.304 | 0.233 | 0.278 | 0.721 | 0.240 | 0.820 | 0.241 | 0.531 | 0.408 |
| RMO4 | 0.333 | 0.275 | 0.334 | 0.748 | 0.323 | 0.852 | 0.302 | 0.621 | 0.456 |
| SCC1 | 0.245 | 0.570 | 0.233 | 0.275 | 0.211 | 0.271 | 0.796 | 0.380 | 0.384 |
| SCC2 | 0.219 | 0.619 | 0.154 | 0.221 | 0.216 | 0.241 | 0.836 | 0.351 | 0.382 |
| SCC3 | 0.208 | 0.582 | 0.246 | 0.255 | 0.229 | 0.285 | 0.850 | 0.369 | 0.369 |
| SCC4 | 0.225 | 0.569 | 0.230 | 0.295 | 0.263 | 0.277 | 0.832 | 0.403 | 0.423 |
| SCR1 | 0.346 | 0.360 | 0.301 | 0.610 | 0.294 | 0.633 | 0.383 | 0.845 | 0.554 |
| SCR2 | 0.331 | 0.331 | 0.342 | 0.532 | 0.283 | 0.559 | 0.369 | 0.863 | 0.535 |
| SCR3 | 0.355 | 0.435 | 0.324 | 0.498 | 0.352 | 0.502 | 0.404 | 0.867 | 0.616 |
| SCR4 | 0.354 | 0.465 | 0.416 | 0.582 | 0.320 | 0.566 | 0.393 | 0.840 | 0.726 |
| SSP1 | 0.335 | 0.445 | 0.280 | 0.466 | 0.318 | 0.457 | 0.420 | 0.633 | 0.882 |
| SSP2 | 0.362 | 0.455 | 0.382 | 0.527 | 0.306 | 0.476 | 0.423 | 0.676 | 0.889 |
| SSP3 | 0.351 | 0.414 | 0.240 | 0.430 | 0.254 | 0.431 | 0.396 | 0.578 | 0.862 |

Prior studies have proposed Heterotrait-monotrait (HTMT) analysis for the testing of discriminant validity. HTMT analysis was introduced by Gold, Malhotra, and Segars [55] and suggests that the HTMT ratios must be lower than 0.85 or 0.90 to indicate the adequate discriminant validity of the constructs [55,56]. The results of the HTMT ratio analysis show

that the HTMT values are less than 0.90 and hence, confirm the adequate discriminant validity of the variables. The values of HTMT are presented in Table 5.

**Table 5.** Heterotrait-Monotrait analysis.

|  | BDA | COM | INT | LCO | NCO | RMO | SCC | SCR | SSP |
|---|---|---|---|---|---|---|---|---|---|
| BDA | | | | | | | | | |
| COM | 0.374 | | | | | | | | |
| INT | 0.637 | 0.318 | | | | | | | |
| LCO | 0.516 | 0.394 | 0.518 | | | | | | |
| NCO | 0.757 | 0.336 | 0.690 | 0.503 | | | | | |
| RMO | 0.519 | 0.398 | 0.536 | 0.170 | 0.460 | | | | |
| SCC | 0.335 | 0.832 | 0.343 | 0.381 | 0.339 | 0.406 | | | |
| SCR | 0.496 | 0.537 | 0.516 | 0.771 | 0.437 | 0.810 | 0.525 | | |
| SSP | 0.494 | 0.585 | 0.446 | 0.656 | 0.406 | 0.651 | 0.553 | 0.821 | |

### 4.2.2. Assessing Structural Model

The structural model tests the hypothesized relationship between variables. In order to mitigate the normality issue, data were bootstrapped with dummy data of 4000, as suggested by [58]. The results of the structural model are given in Table 6 comprising path value, standard error, t-statistics and the significance of the hypotheses.

**Table 6.** Results of the hypotheses.

| Hypothesis | Relationship | Path Coefficient | STDEV | T-Statistics | Significance | Decision |
|---|---|---|---|---|---|---|
| H1 | INT $\rightarrow$ SCR | 0.113 | 0.043 | 20.634 | 0.005 | Accepted |
| H2 | COM $\rightarrow$ SCR | 0.183 | 0.064 | 20.859 | 0.003 | Accepted |
| H3 | LCO $\rightarrow$ SCR | 0.210 | 0.080 | 20.635 | 0.005 | Accepted |
| H4 | RMO $\rightarrow$ SCR | 0.326 | 0.081 | 40.012 | 0.000 | Accepted |
| H5 | SCC $\rightarrow$ SCR | 0.116 | 0.060 | 10.933 | 0.028 | Accepted |
| H6 | NCO $\rightarrow$ SCR | 0.028 | 0.042 | 0.679 | 0.249 | Not Accepted |
| H7 | SCR $\rightarrow$ SSP | 0.667 | 0.032 | 20.678 | 0.000 | Accepted |

The results of the structural model indicate that although the exogenous variables have a significant impact on supply chain resilience, the relationship between network capacity and supply chain resilience is insignificant. The following results indicate that supply chain intelligence significantly impacts supply chain resilience and support H1: $\beta = 0.113$ path; significance, $p < 0.001$ and t-statistics, 2.634. Supply chain communication has a positive impact on supply chain resilience and confirms H2: $\beta = 0.183$ path; significance, $p < 0.001$ and t-statistics, 2.859. Similarly, leadership commitment has revealed a positive impact on measuring supply chain resilience and statistically confirms H3: $\beta = 0.210$ path; significance, $p < 0.001$ and t-statistics, 2.635. The statistics show that both risk management orientation and supply chain capability have a positive impact on supply chain resilience and are supported by $\beta = 0.326$ path; significance, $p < 0.001$ and t-statistics, 4.012; $\beta = 0.116$ path, significance, $p < 0.05$ and t-statistics, 1.933, hence establishing H4 and H5. Contrary to the researcher, the relationship between the expected network complexity and supply chain resilience was found to be insignificant ($\beta = 0.028$ path; significance, $p > 0.05$ and t-statistics, 0.679) and, therefore, H6 is rejected. Finally, the supply chain resilience has shown a positive impact toward sustainable supply chain performance and confirms H7: $\beta = 0.667$ path; significance, $p < 0.001$ and t-statistics, 2.678, the results of hypotheses including path coefficient and significant level are shown in Appendix A.

### 4.2.3. Assessing Effect Size, Predictive Power and Coefficient of Determination

The effect size of the variables is examined through $f^2$ analysis [50,51]. According to Samar Rahi et al. [58] the coefficient of determination, $R^2$, reveals the collective impact of

all the exogenous variables towards the criterion variable. Nevertheless, variable size as a single factor should be assessed with effect size analysis. The results demonstrate that in measuring the supply chain sustainable performance, the effect size of supply chain resilience is substantial and, therefore, a potential construct for managerial implication.

Concerning the coefficient of determination, the results of the structural model revealed that exogenous variables have a substantial variance $R^2$ of 0.548% on supply chain resilience. Similarly, the findings indicate a substantial variance $R^2$ of 0.550% in measuring sustainable supply chain performance, which was predicted by supply chain resilience and big data analytics. Aside from the substantial coefficient of determination for supply chain resilience and sustainable supply chain performance, the predictive power $Q^2$ of the framework was computed. The result of the blindfolding method has established that the research model has substantial power to predict supply chain resilience and sustainable supply chain performance. The results of the effect size, predictive power and coefficient of determination are shown in Table 7.

**Table 7.** Measuring $R^2$, $f^2$ and $Q^2$.

| Supply Chain Resilience | | | | |
| --- | --- | --- | --- | --- |
| Constructs | $R^2$ | $Q^2$ | $f^2$ | Findings |
| Supply Chain Resilience | 54.8% | 37.4% | | |
| Supply chain communication | | | 0.036 | Small |
| Supply chain intelligence | | | 0.019 | Small |
| Leadership commitment | | | 0.018 | Small |
| Network complexity | | | 0.001 | Small |
| Risk management orientation | | | 0.045 | Small |
| Supply chain capability | | | 0.014 | Small |
| Sustainable Supply Chain Performance | | | | |
| Constructs | $R^2$ | $Q^2$ | $f^2$ | Findings |
| Sustainable Supply Chain Performance | 55.0% | 40.0% | | |
| Big data analytics | | | 0.030 | Small |
| Supply chain resilience | | | 0.826 | Substantial |

4.2.4. Importance and Performance Analysis

The current research model has integrated numerous factors to determine sustainable supply chain performance. Therefore, the importance and performance of the variable are tested with importance performance matrix analysis (IPMA) consistent with earlier studies by [42,59]. Before applying IPMA analysis, it was important to choose an outcome variable. In this study, the researcher selected supply chain sustainable performance as the outcome variable. The findings of the importance performance analysis indicate that supply chain resilience has the highest importance/total effect in measuring sustainable supply chain performance. Therefore, the importance of big data analytics, risk management orientation, supply chain communication and leadership commitment show an intermediate level of importance. The results of the IPMA analysis are exhibited in Table 8 with the importance and performance indexes.

The importance of the constructs is observed through an IPMA map. The IPMA map exhibits that supply chain capability, intelligence and network complexity have less importance in measuring sustainable supply chain performance. Therefore, the importance of big data analytics, risk management orientation, supply chain communication, leadership commitment and supply chain resilience is considerable. Thus, managers and policy makers should focus on factors such as the importance of big data analytics, risk management orientation, supply chain communication, leadership commitment and supply chain resilience to enhance the sustainable supply chain performance in their organization. The IPMA map exhibited in Figure 2 clearly shows the performance of the constructs on the *Y*-axis and importance on the *X*-axis.

**Table 8.** Results of IPMA.

| Constructs | Total Effects of Constructs | Total Performance of the Constructs |
|---|---|---|
| Big data analytics | 0.157 | 7.890 |
| Supply chain communication | 0.128 | 73.379 |
| Supply chain intelligence | 0.095 | 69.381 |
| Leadership commitment | 0.167 | 67.870 |
| Network complexity | 0.023 | 72.166 |
| Risk management orientation | 0.261 | 68.990 |
| Supply chain capability | 0.086 | 71.953 |
| Supply chain resilience | 0.690 | 66.904 |

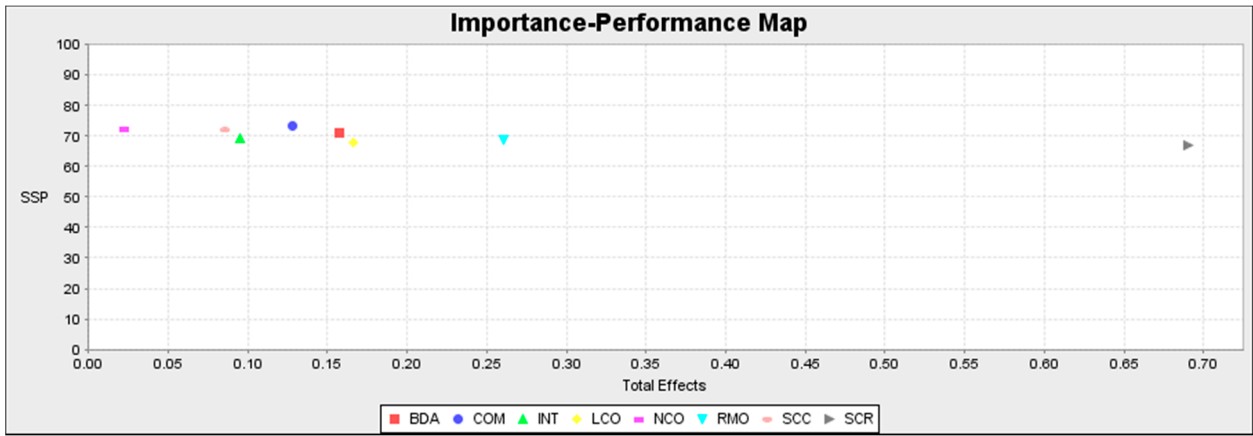

**Figure 2.** IPMA analysis map.

### 4.3. Moderating Effect of Big Data Analytics

The impact of technology is substantial in achieving the strategic goals of a firm [60]. In line with the above argument, this study has outlined big data analytics in a research model as moderating the variables between supply chain resilience and sustainable supply chain performance. The moderating effect of big data analytics was tested through a product indicator approach consistent with earlier studies by [53]. For statistical computation, the orthogonalization method was selected. The findings indicate that BD analytics moderates the relationship outlined between supply chain resilience and sustainable supply chain performance and statistically confirms H8 ($\beta$ = 0.138; significant at $p < 0.001$; t-statistics, 4.094). Figure 3 exhibits the findings of the moderating effect.

The moderating effect was further analyzed with simple slope analysis to test the strength of the moderating effect. According to S. Rahi [61] simple slope analysis reveals the trend of the moderating effect i.e., whether it moderates strongly or weakly. Nevertheless, the result of the simple slope analysis shows an upward BDA trend of +1SD on the green line compared to the negative BDA at −SD on the red line, which means that simple slope analysis has a declining trend. Nonetheless, the blue line indicates a neutral impact between the highest and lowest moderating effect. Hence, the result of the moderating effect establishes that the higher use of big data analytics in a supply chain operation will increase the supply chain resilience and sustainable supply chain performance. The simple slope analysis graph is displayed in Figure 4.

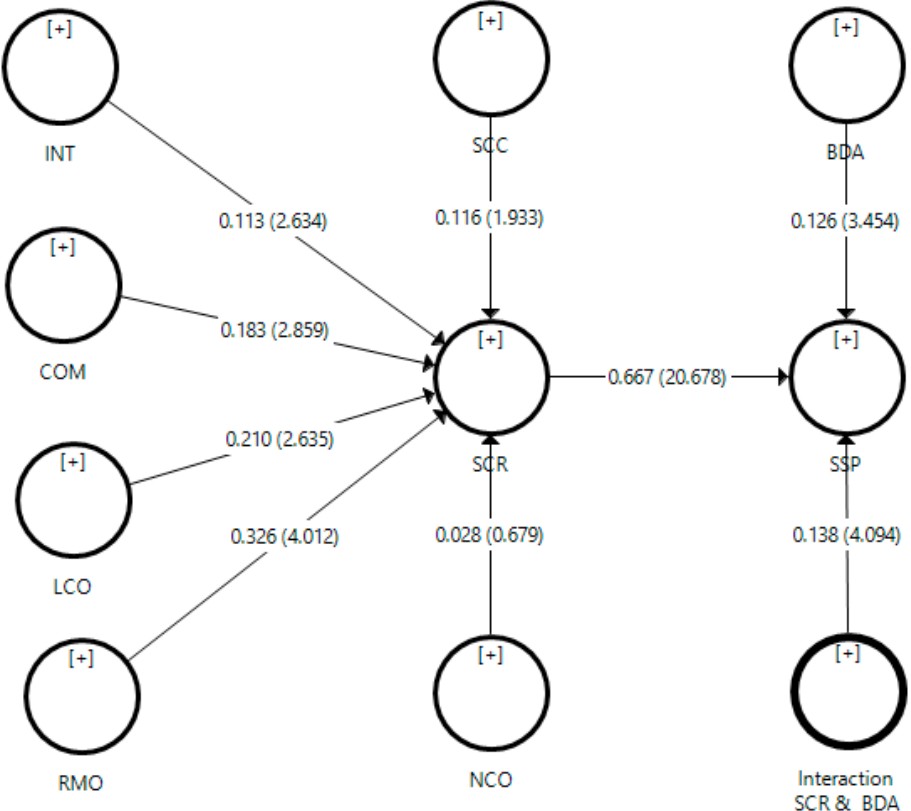

**Figure 3.** Moderating analysis.

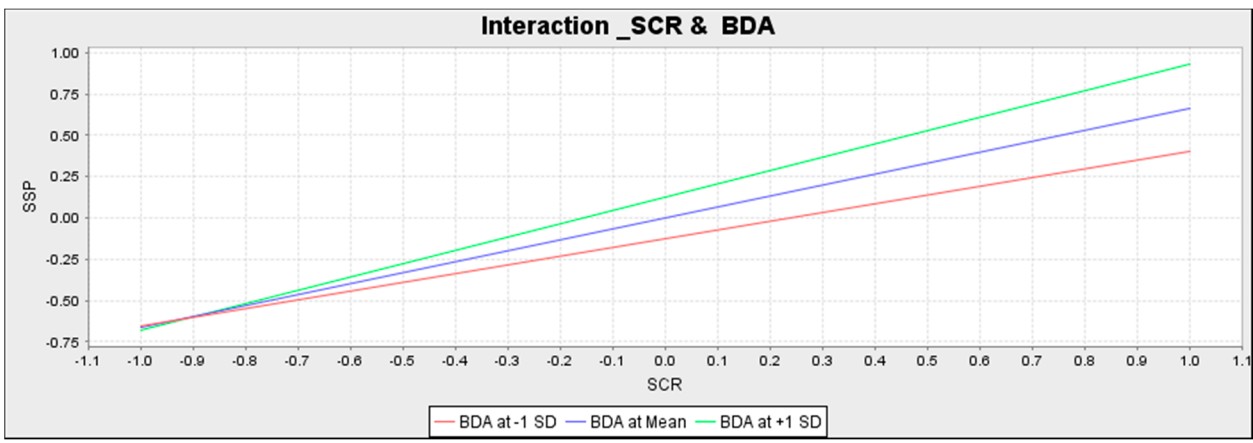

**Figure 4.** Output of simple slope analysis.

## 5. Discussion

The results of this research unfold interesting facts for both academic researchers and practitioners. The current research has synthesized the literature into two main streams. At the first stage, supply chain resilience is determined by supply chain intelligence, supply chain communication, leadership commitment, risk management orientation, supply chain capability, network complexity and the substantial variance $R^2$ of 54.8%. Therefore, the second stream of the literature focused on sustainable supply chain performance with supply chain resilience and big data analytics and revealed a considerable variance $R^2$ of 55.0% in sustainable supply chain performance. These findings established the theoretical validity of the research model in determining supply chain resilience and sustainability

supply chain performance. Similarly, the literature has confirmed the moderating role of big data analytics between the relationship of supply chain resilience and sustainable supply chain performance. The moderating effect of big data analytics indicates that the higher use of BD analytics in supply chain operations will raise supply chain resilience and sustainable supply chain performance.

Concerning the hypothesized relationships, the findings of the structural model indicate that supply chain intelligence positively influences supply chain resilience, which is consistent with earlier studies by [11,18]. Similarly, supply chain communication has shown a positive impact on supply chain resilience, which is in line with [12,22,38]. Leadership commitment had revealed a positive impact on measuring supply chain resilience, which supports prior studies by [13,15]. Pointing to risk management orientation and supply chain capability, both factors have shown a positive impact on supply chain resilience, which was supported by earlier studies by [28–30]. Nevertheless, network complexity has shown an insignificant influence supply chain resilience that was beyond our expectations. Therefore, supply chain resilience has shown a positive impact toward sustainable supply chain performance, which is consistent with an earlier study by [17]. Following IPMA and effect size analysis, this research suggests that during the COVID-19 pandemic, supply chain resilience and sustainable supply chain process could be enhanced by focusing on factors such as big data analytics, risk management orientation, supply chain communication and leadership commitment.

*Research Contribution to Theory and Practice*

The findings of this research have contributed to theory and practice. For instance, the current research shows that factors such as supply chain intelligence, communication, leadership commitment, risk orientation, supply chain capability and network complexity positively relate to supply chain resilience. These findings clearly indicate that policy makers and supply chain managers should focus on the outlined factors to enhance supply chain resilience in supply chain operations. Another aspect of this research is to shed light on sustainable supply chain performance. The results of this study reveal that a sustainable supply chain is determined by supply chain resilience and big data analytics. Therefore, supply chain managers should improve supply chain resilience in order to improve sustainable supply chain performance. Aside from industry utilization, this research contributes to the academic literature by developing an integrated supply chain model that comprises market-oriented factors to determine supply chain resilience and sustainable supply chain performance. In addition to that, the variance explained by exogenous variables was substantial for supply chain resilience and sustainable supply chain performance, which, in turn, confirms the validity of the research model. Furthermore, this study contributes to information system literature by testing the moderating effect of big data analytics between supply chain resilience and sustainable supply chain performance. The findings establish that a higher use of big data analytics in supply chain operations will increase supply chain resilience and sustainable supply chain performance. Therefore, managers and policy makers should introduce big data analytics tools in supply chain operations to boost supply chain resilience and sustainable supply chain performance.

## 6. Conclusions

The aim of this study was to investigate factors that influence supply chain resilience and sustainable supply chain performance during the COVID-19 pandemic. The current study developed an integrated research model that combined factors such as supply chain intelligence, supply chain communication, leadership commitment, risk management orientation, supply chain capability and network complexity to investigate supply chain resilience. On the other side, the research model was extended with the moderating effect of big data analytics between the relationship of supply chain resilience and sustainable supply chain performance. The research model of this study was tested using structural equation modeling. The results of the structural model computation revealed that supply

chain intelligence, supply chain communication, leadership commitment, risk management orientation, supply chain capability and network complexity have shown a substantial variance $R^2$ of 0.548% toward supply chain resilience. Therefore, supply chain resilience and big data analytics have an explanation to the variance $R^2$ of 0.550% in measuring sustainable supply chain performance. In addition to that, the validity of the research model was tested with predictive analysis $Q^2$ using a blind folding procedure. The results of the predictive analysis $Q^2$ revealed that the research model has large predictive power to predict supply chain resilience and sustainable supply chain performance 37.4% and 40.0%, respectively. This research contributes to the theory by developing an integrated research model toward supply chain resilience and sustainable supply chain performance. Therefore, for practical implications, this study suggests that supply chain managers should focus on factors such as big data analytics, risk management orientation, supply chain communication and leadership commitment to enhance supply chain resilience and sustainable supply chain performance. In addition to that, the moderating effect of big data analytics is confirmed between the relationship of supply chain resilience and sustainable supply chain performance. These findings established that the use of big data analytics in a supply chain operation will increase supply chain resilience and sustainable supply chain performance. Thus, supply chain managers and policy makers should incorporate big data analytics in supply chain operations to enhance supply chain resilience and sustainable supply chain performance.

*Research Limitations and Future Direction*

This research has some limitations that reveal a potential area of future research. First, this study was developed as an integrated supply chain research model that combined factors such as supply chain intelligence, supply chain communication, leadership commitment, risk management orientation, supply chain capability, network complexity and big data analytics to investigate supply chain resilience and sustainable supply chain performance phenomenon. Nevertheless, there are some other factors that could impact supply chain resilience such as supplier relationship, uncertainty and inter departmental coordination. Thus, extending the current research model with some additional factors could reveal interesting findings. Another limitation of this research is related to research design. The research design was based on a cross-sectional design and, therefore, respondents were restrained to participate at once in the research survey. It is expected that results may differ in a longitudinal research design. Therefore, the supply chain resilience phenomenon should be investigated with a longitudinal research design. The data were collected through a convenience sampling approach, which is a non-probability sampling approach. It is suggested that future researchers collect data through a probability sampling approach to mitigate any kind of sampling risk. The research model was designed from a developing countries perspective and tested the resilience behavior of a Saudi manufacturer. However, a future researcher may test the current research model in the context of other developing countries to enhance the generalizability of the research model.

**Funding:** This work is funded by the University of Jeddah, Jeddah, Saudi Arabia, under grant No. (UJ-21-DR-46). The authors, therefore, acknowledge with thanks the University of Jeddah technical and financial support.

**Institutional Review Board Statement:** Not Applicable.

**Informed Consent Statement:** Not Applicable.

**Data Availability Statement:** Delivered on demand.

**Acknowledgments:** Author would like to thank anonymous reviewers for the critical and constructive suggestions which in turn give an opportunity to rethink and reconstruct manuscript. In addition to that, author is grateful to the editor-in-chief and assistant editor for their humbleness during the revision process.

**Conflicts of Interest:** The author declares no conflict of interest. The authors declare that they have no known competing financial interests or personal relationships that could have appeared to influence the work reported in this paper.

**Appendix A. Path Coefficient and Significance Level**

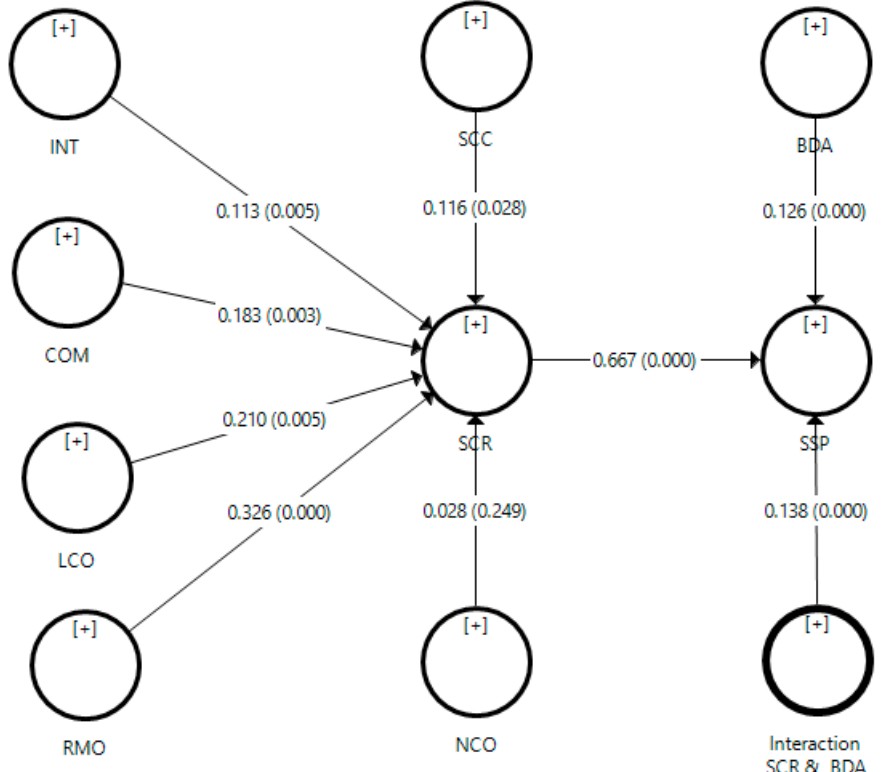

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
