# Peer review of "Investigating the Drivers of Supply Chain Resilience in the Wake of the COVID-19 Pandemic: Empirical Evidence from an Emerging Economy"

_sustainability, doi:10.3390/su132111939_

Round 1
Reviewer 1 Report
Interesting paper and good materials.
Abstract:
The originality and the significance of the research paper should be highlighted in the abstract.
Introduction:
the last paragraph should explain the remaining structure of the paper
Literature Review:
More clarifications and review of the literature for sections 2.1. Supply chain intelligence and communication, 2.2. The role of leadership commitment and risk management orientation, 2.3. Supply chain capability and network complexity, 2.4. Big data analytics are needed to show that Authors comprehensively reviewed the literature.
Solution Methodology:
Sounds fine
Results and Discussion:
There is a lack of clarity within the discussion and findings section. A more precise analysis is required and justify the results by supporting via references from previous findings where applicable. 2 subsections for theoretical and managerial contributions are needed.
Conclusion:
Sounds fine
Overall, the work is fine and by addressing the above-mentioned comments, I support its publication.
Author Response
Reviewer Comments to Author & Author response to Reviewer
Respected Reviewers/Editor
With reference to Manuscript “Investigating the Drivers of Supply Chain Resilience in the Wake of COVID-19 Pandemic: Empirical Evidence from Emerging Economy", sustainability-1388675, author has answered reviewer comments in following section.
Dear Editor and Team,
Greetings,
Thanks for your suggestion regarding manuscript entitled " Investigating the Drivers of Supply Chain Resilience in the Wake of COVID-19 Pandemic: Empirical Evidence from Emerging Economy "
Suggestion from Editor and reviewers are fruitful to improve this paper. Therefore, researcher has carefully revised the manuscript and changed it according to reviewer’s recommendations. Additionally, literature is also updated with latest studies.
I hope that the changes I have made resolve all your concerns about the manuscript. I am more than happy to make any further changes that will improve the paper and/or facilitate successful publication.
Thank you once again for your time and interest. I look forward to hearing from you.
Reviewer 1 Comments to Author and Author Response
Reviewer Comments:
English language and style are fine/minor spell check required
Author Response:
Regarding your concern, author has reviewed contents carefully and resolved all issues. However, there are some scientific words such as Heterotrait monotrait (HTMT) and Fornell and Larcker analysis that cannot be changed.
Reviewer Comments to different sections
Abstract:
The originality and the significance of the research paper should be highlighted in the abstract.
Author Response:
Yes, usefulness of this study is highlighted in practical context, this study suggests that supply chain managers should focus on factors such as big data analytics, risk management orientation, supply chain communication and leadership commitment to enhance supply chain resilience and sustainable supply chain performance. In addition to that initial sentence abstract demonstrates the need of this study. “The COVID-19 pandemic has disrupted supply chain operations globally. Nevertheless, resilient firms have capacity to combat with unprecedented situation with right strategic approach”.
Introduction:
the last paragraph should explain the remaining structure of the paper
Author Response:
As per your suggestion remaining section detail is added. The remaining part of the research explains literature review, methodology, data analysis, discussion and conclusion of this study.
Literature Review:
More clarifications and review of the literature for sections 2.1.Supply chain intelligence and communication, 2.2. The role of leadership commitment and risk management orientation, 2.3. Supply chain capability and network complexity, 2.4. Big data analytics are needed to show that Authors comprehensively reviewed the literature.
Author Response:
Dear reviewer, literature is updated with latest citation and clear deflations of constructs. Author has updated supply chain intelligence and communication section in red. Similarly, other sections are updated including the role of leadership commitment and risk management orientation, supply chain capability and network complexity and big data analytics.
Solution Methodology:
Sounds fine
Author Response:
Thank you so much for appreciation.
Results and Discussion:
There is a lack of clarity within the discussion and findings section. A more precise analysis is required and justify the results by supporting via references from previous findings where applicable.2 subsections for theoretical and managerial contributions are needed.
Author Response:
In discussion section, first precise information is given about results, then findings of this research are compared and contrast with prior studies. Concerning with implications, author has separately divided this section with 5.1 and explains contribution of this research both to theory and practice.
Conclusion:
Sounds fine
Author Response:
Thank you so much for appreciation.
Overall, the work is fine and by addressing the above-mentioned comments, I support its publication.
Author Response:
Thank you so much, author has carefully reviewed and changes are made in manuscript accordingly.

Reviewer 2 Report
The paper has some merits but
- Average variance extracted (AVE) is not reliable for structural equation modelling. See Henseler, J., Ringle, C. M., Sarstedt, M., 2014. A new criterion for assessing discriminant validity in variance-based structural equation modeling. Journal of the Academy of Marketing Science 43 (1), 115–135) Therefore the author should prove his case
- PLS can be applied as an alternative to multiple linear regression (MLR) because the latter fails “when the factors are few in number, are not significantly redundant (collinear), and have a well-understood relationship to the responses, then can be a good way to turn data into information)“ Therefore the authors should show that this is the case Tobias, R. D. (1995, April). An introduction to partial least squares regression. In Proceedings of the twentieth annual SAS users group international conference (Vol. 20). Cary: SAS Institute Inc.
- Regarding effect size the author may refer to Cohen's term d and show the result. See Cohen, Jacob. "Things I have learned (so far)." In Annual Convention of the American Psychological Association, 98th, Aug, 1990, Boston, MA, US; Presented at the aforementioned conference. American Psychological Association, 1992. Ellis, P. D. (2010). The essential guide to effect sizes: Statistical power, meta-analysis, and the interpretation of research results. Cambridge university press.
- Regarding the case in which the regressand is not a continuous variable but a state, which may or may not hold, or a category in a given classification the author should mention why he does not use logit analysis. See Cramer, J. S. (2003). Logit models from economics and other fields. Cambridge University Press. Orlando, G., & Pelosi, R. (2020). Non-Performing Loans for Italian Companies: When Time Matters. An Empirical Research on Estimating Probability to Default and Loss Given Default. International Journal of Financial Studies, 8(4), 68.
- The layout is wrong, for example figures and tables are bigger than text width
- Few spelling mistakes, for example “commination channels” perhaps should be “communication channels”
- The paper seems to reference disproportionately some authors that are not very close to get a Nobel prize, for example Rahi is cited 10 times. The author should review carefully the bibliography
Author Response
Reviewer Comments to Author & Author response to Reviewer
Respected Reviewers/Editor
With reference to Manuscript “Investigating the Drivers of Supply Chain Resilience in the Wake of COVID-19 Pandemic: Empirical Evidence from Emerging Economy", sustainability-1388675, author has answered reviewer comments in following section.
Dear Editor and Team,
Greetings,
Thanks for your suggestion regarding manuscript entitled " Investigating the Drivers of Supply Chain Resilience in the Wake of COVID-19 Pandemic: Empirical Evidence from Emerging Economy "
Suggestion from Editor and reviewers are fruitful to improve this paper. Therefore, researcher has carefully revised the manuscript and changed it according to reviewer’s recommendations. Additionally, literature is also updated with latest studies.
I hope that the changes I have made resolve all your concerns about the manuscript. I am more than happy to make any further changes that will improve the paper and/or facilitate successful publication.
Thank you once again for your time and interest. I look forward to hearing from you.
Reviewer 2 Comments to Author and Author response to reviewer
Reviewer Comments:
1 Average variance extracted (AVE) is not reliable for structural equation modelling. See Henseler, J., Ringle, C. M., Sarstedt, M., 2014. A new criterion for assessing discriminant validity in variance-based structural equation modeling. Journal of the Academy of Marketing Science 43 (1), 115–135) Therefore the author should prove his case.
Author Response:
Dear reviewer Average variance extracted is used in this research to measure convergent validity as shown in Table 2. Therefore, square root of AVE is used to measure discriminant validity. Concerning with new method provided by Henseler, J., Ringle, C. M., & Sarstedt, M. (2015), author has done reading and assessed discriminant validity through Heterotrait monotrait (HTMT) analysis.
Henseler, J., Ringle, C. M., & Sarstedt, M. (2015). A new criterion for assessing discriminant validity in variance-based structural equation modeling. Journal of the Academy of Marketing Science, 43(1), 115-135.
The results of the HTMT analysis are shown in Table 5.
Reviewer Comments:
- PLS can be applied as an alternative to multiple linear regression (MLR) because the latter fails “when the factors are few in number, are not significantly redundant (collinear), and have a well-understood relationship to the responses, then can be a good way to turn data into information)“ Therefore the authors should show that this is the case Tobias, R. D. (1995, April). An introduction to partial least squares regression. In Proceedings of the twentieth annual SAS users group
Author Response:
Dear Reviewer, the partial least square approach is applied in line with F. Hair Jr, J., Sarstedt, M., Hopkins, L., &G. Kuppelwieser, V. (2014) suggested that PLS approach is appropriate to be taken into data analysis wherein the objective of research is towards theory developing instead of theory testing.
- Hair Jr, J., Sarstedt, M., Hopkins, L., & G. Kuppelwieser, V. (2014). Partial least squares structural equation modeling (PLS-SEM) An emerging tool in business research. European Business Review, 26(2), 106-121.
Further support is taken from Rahi, S. (2017). Structural Equation Modeling Using SmartPLS. CreateSpace Independent Publishing Platform stated that PLS approach appropriate in less matured model. As this research model is an integrated model that combines several factors hence, author has selected VB-SEM to conduct data analysis.
Reviewer Comments:
- Regarding effect size the author may refer to Cohen's term d and show the result. See Cohen, Jacob. "Things I have learned (so far)." In Annual Convention of the American Psychological Association, 98th, Aug, 1990, Boston, MA, US; Presented at the aforementioned conference.
American Psychological Association, 1992. Ellis, P. D. (2010). The essential guide to effect sizes: Statistical power, meta-analysis, and theinterpretation of research results. Cambridge university press
Author Response:
Dear reviewer for effect size analysis suggested reference is added. Cohen, J. (1992). Things I have learned (so far). Annual Convention of the American Psychological Association, 98th, Aug, 1990, Boston, MA, US; Presented at the aforementioned conference.
Reviewer Comments:
- Regarding the case in which the regressand is not a continuous variable but a state, which may or may not hold, or a category in a given classification the author should mention why he does not use logit analysis. See Cramer, J. S. (2003). Logit models from economics and other fields. Cambridge University Press.
Orlando, G., & Pelosi, R. (2020). Non-Performing Loans for Italian Companies: When Time Matters. An Empirical Research on Estimating Probability to Default and Loss Given Default. International Journal of Financial Studies, 8(4), 68.
Author Response:
This research is behavior and therefore test causal relationship among variables. Overall positivist paradigm is followed as suggested by Rahi, S. (2017).
Rahi, S. (2017). Research design and methods: A systematic review of research paradigms, sampling issues and instruments development. International Journal of Economics & Management Sciences, 6(2), 1-5.
In addition to that researcher has used some additional test to see predictive power of the research model such as blindfolding analysis using consistent with
Geisser, S. (1974). A predictive approach to the random effect model. Biometrika, 61(1), 101-107.
Reviewer Comments:
5 The layout is wrong, for example figures and tables are bigger
than text width
Author Response:
Copyediting of this manuscript is yet to be done. Problem occurs due to template and author has requested to editor for assistance. In final version, this issue will be resolved.
Reviewer Comments:
- Few spelling mistakes, for example “commination channels” perhaps should be “communication channels”
Author Response:
Author has resolved this issue and highlighted in text body.
Reviewer Comments:
- The paper seems to reference disproportionately some authors that are not very close to get a Nobel prize, for example Rahi is cited 10 times. The author should review carefully the bibliography.
Author Response: The only objective to cite highlighted author is to improve methodology section with latest research. Author found useful methods from Rahi et al and hence citied multiple time. At the same time all references are latest not old dated

Reviewer 3 Report
The contribution is generally accurate. However, a careful rereading of the text is suggested in order to:
- correct some inconsistencies that have remained (e.g. at line n. 89: “A recent study conducted by Asamoah et al. (2021) has confirmed that both communication and supply chain resilience positively impact supply chain resilience”; I suppose that is: “… communication and supply chain intelligence positively impact supply chain resilience”),
- avoid leaving the titles of the paragraphs at the bottom of the page (e.g. at lines 94, p. 2, and 218, p. 5),
- standardize bibliographic references; particularly:
- at line 100 “Mohammed Ali Yousef Yamin” is Mohammad, furthermore the name of the authors it is no needed! So, at least the following lines must be corrected:
- line 100
- line 180
- line 213
- line 215
- line 216
- at line 100 “Mohammed Ali Yousef Yamin” is Mohammad, furthermore the name of the authors it is no needed! So, at least the following lines must be corrected:
and line 573 adding Yamin, M.A.Y.
- similarly, Majeed Mustafa Othman Mansour has to be cited in the text as “Othman Mansour” and in the Reference “Othman Mansour M.M.”; so, at least the following lines must be corrected:
- line 213
- line 284 (n. 2)
- line 547
- it is necessary to differently recall the different citations in the text; e.g.:
- Rahi, S. (2017a) and Rahi, S. (2017b)
- Yamin, M.A.Y. (2020a), Yamin, M.A.Y. (2020b) and Yamin, M.A.Y. (2020c)
- and so on…
Author Response
Reviewer Comments to Author & Author response to Reviewer
Respected Reviewers/Editor
With reference to Manuscript “Investigating the Drivers of Supply Chain Resilience in the Wake of COVID-19 Pandemic: Empirical Evidence from Emerging Economy", sustainability-1388675, author has answered reviewer comments in following section.
Dear Editor and Team,
Greetings,
Thanks for your suggestion regarding manuscript entitled " Investigating the Drivers of Supply Chain Resilience in the Wake of COVID-19 Pandemic: Empirical Evidence from Emerging Economy "
Suggestion from Editor and reviewers are fruitful to improve this paper. Therefore, researcher has carefully revised the manuscript and changed it according to reviewer’s recommendations. Additionally, literature is also updated with latest studies.
I hope that the changes I have made resolve all your concerns about the manuscript. I am more than happy to make any further changes that will improve the paper and/or facilitate successful publication.
Thank you once again for your time and interest. I look forward to hearing from you.
Comments from Reviewers 3
Reviewer Comments:
- correct some inconsistencies that have remained (e.g. at line n.89: “A recent study conducted by Asamoah et al. (2021) hasconfirmed that both communication and supply chain resilience positively impact supply chain resilience”; I supposethat is: “… communication and supply chain intelligence positively impact supply chain resilience”),
Author Response: Issue is resolve and thank you so much for highlighting.
Reviewer Comments:
- avoid leaving the titles of the paragraphs at the bottom of the page (e.g. at lines 94, p. 2, and 218, p. 5),
Author Response: Issue addressed as heading of the paragraph is moved to page 3 “The role of leadership commitment and risk management orientation”
Reviewer Comments:
- standardize bibliographic references; particularly: at line 100 “
Mohammed Ali Yousef Yamin” is Mohammad, furthermore the name of the authors it is no needed! So, at least the following lines must be corrected:
and line 573 adding Yamin, M.A.Y. similarly, Majeed Mustafa Othman Mansour has to be cited in the text as “Othman Mansour” and in the Reference “Othman MansourM.M.”; so, at least the following lines must be corrected:
it is necessary to differently recall the different citations in the text;e.g.:Rahi, S. (2017a) and Rahi, S. (2017b) Yamin, M.A.Y. (2020a), Yamin, M.A.Y. (2020b) and Yamin,M.A.Y. (2020c)
and so on…
Author Response: Issued addressed and highlighted in text in addition to that author has used End Note software for citations and hopefully citation would be appropriate. Regarding style, Author has used APA 7th as APA 6th is no more existed.

Reviewer 4 Report
Dear Author,
Thanks for your submission to Sustainability. The ideas of the article looks fine. That being said, I have the following comments.
- Where the constructs and associated indicators/items are taken from? Show a table to justify them?
- What are the new constructs and items in context of COVID-19? It is crucial to know. Are all previous constructs and items adequate to face the pandemic.
- All hypotheses are taken from findings of other studies? Or, those are derived using arguments of previous studies? Is there any relation among constructs and hypotheses in respect of past studies?
- In the Appendix or in the supplementary materials, provide the structural equations, if possible, as the model is a structural equation model.
- The article must ass implications to sustainability or sustainable development goals. Check the below articles to gain ideas on it. Also this work misses many key articles on COVID-19 and sustainability, even I see it misses relevant articles published in Sustainability journal. For example, Strategies to Manage the Impacts of the COVID-19 Pandemic in the Supply Chain: Implications for Improving Economic and Social Sustainability, published in Sustainability, and Challenges to COVID-19 vaccine supply chain: Implications for sustainable development goals, published in IJPE.
- Conclusion seems lengthy. Concise them without repeating information already said before.
- Present the summary of demographic profile of the respondents.
- What happens if I want to add or remove an item from a construct?
Author Response
Reviewer Comments to Author & Author response to Reviewer 5
Respected Reviewers
With reference to Manuscript “Investigating the Drivers of Supply Chain Resilience in the Wake of COVID-19 Pandemic: Empirical Evidence from Emerging Economy" author has reviewed article as per your suggestion.
Reviewer Comments
- Where the constructs and associated indicators/items are taken from? Show a table to justify them?
1 Author Response: Dear reviewer information is given in section 3.1 namely Designing questionnaire and instrument development. Further detail is as follows
Scale items for network complexity were adopted from Durach et al. (2015). Risk management orientation scale is adopted from Wieland and Marcus Wallenburg (2012) and then slightly adapted into current research context. Supply chain resilience scale items are adopted from previously developed scale by Brandon‐Jones et al. (2014) then slightly adapted. Scale items for the construct big data analytics were adopted from Dubey et al. (2021). Supply chain communication scale items were adopted from Zhang and Cao (2018).Similarly, supply chain intelligence items were adopted from Asamoah et al. (2021). Therefore, supply chain capability scale is adapted from Wu et al. (2006) and Asamoah et al. (2021). Scale items for leadership commitment were adapted from Kaynak (2003). Construct items for sustainable supply chain performance were adopted from Gunasekaran et al. (2017) and Bag, Wood, Xu, Dhamija, and Kayikci (2020).
Reviewer Comments
- What are the new constructs and items in context of COVID-19? It is crucial to know. Are all previous constructs and items adequate to face the pandemic.
- Author Comments: Dear reviewer in this research new scale is not developed as the objective of this research is to develop causal relationship among exogenous and endogenous factors instead of scale development. All scale items were adopted and then adapted into current research context. Validity, reliability has been checked in measurement model. Please see Table1.
Reviewer Comments
- All hypotheses are taken from findings of other studies? Or, those are derived using arguments of previous studies? Is thereany relation among constructs and hypotheses in respect ofpast studies?
3 Author Response: Dear Reviewer for hypotheses development detailed literature review is conducted and later on arguments developed to interlink relationships.
Reviewer Comments
- In the Appendix or in the supplementary materials, provide thestructural equations, if possible, as the model is a structuralequation model.
- Author Response: Dear Reviewer in Appendix the diagram is output of Structural equation modeling comprising path coefficient values and significance of the path.
Reviewer Comments
- The article must ass implications to sustainability orsustainable development goals. Check the below articles togain ideas on it. Also this work misses many key articles onCOVID-19 and sustainability, even I see it misses relevantarticles published in Sustainability journal. For example,Strategies to Manage the Impacts of the COVID-19 Pandemicin the Supply Chain: Implications for Improving Economic andSocial Sustainability, published in Sustainability, andChallenges to COVID-19 vaccine supply chain: Implications forsustainable development goals, published in IJPE.
- Author Response: Dear reviewer several latest research studied have been added in this research and some references are as follows.
Taqi, H. M., Ahmed, H. N., Paul, S., Garshasbi, M., Ali, S. M., Kabir, G., & Paul, S. K. (2020). Strategies to manage the impacts of the COVID-19 pandemic in the supply chain: implications for improving economic and social sustainability. Sustainability, 12(22), 9483.
Karmaker, C. L., Ahmed, T., Ahmed, S., Ali, S. M., Moktadir, M. A., & Kabir, G. (2021). Improving supply chain sustainability in the context of COVID-19 pandemic in an emerging economy: Exploring drivers using an integrated model. Sustainable production and consumption, 26, 411-427.
Lopes de Sousa Jabbour, A. B., Chiappetta Jabbour, C. J., Hingley, M., Vilalta-Perdomo, E. L., Ramsden, G., & Twigg, D. (2020). Sustainability of supply chains in the wake of the coronavirus (COVID-19/SARS-CoV-2) pandemic: lessons and trends. Modern Supply Chain Research and Applications, 2(3), 117-122. doi: 10.1108/mscra-05-2020-0011
Rahi, S., Khan, M. M., & Alghizzawi, M. (2021). Factors influencing the adoption of telemedicine health services during COVID-19 pandemic crisis: an integrative research model. Enterprise Information Systems, 15(6), 769-793. doi: 10.1080/17517575.2020.1850872
Paul, S. K., Chowdhury, P., Moktadir, M. A., & Lau, K. H. (2021). Supply chain recovery challenges in the wake of COVID-19 pandemic. Journal of Business Research, 136, 316-329.
Belhadi, A., Kamble, S., Jabbour, C. J. C., Gunasekaran, A., Ndubisi, N. O., & Venkatesh, M. (2021). Manufacturing and service supply chain resilience to the COVID-19 outbreak: Lessons learned from the automobile and airline industries. Technological Forecasting and Social Change, 163, 120447.
Butt, A. S. (2021). Strategies to mitigate the impact of COVID-19 on supply chain disruptions: a multiple case analysis of buyers and distributors. The International Journal of Logistics Management, ahead-of-print(ahead-of-print). doi: 10.1108/ijlm-11-2020-0455
Reviewer Comments
- Conclusion seems lengthy. Concise them without repeating information already said before.
6 Author Response: Author has tried his best to reduce conclusion, only relevant findings have been presented in conclusion.
Reviewer Comments
- Present the summary of demographic profile of the respondents.
7 Author response: Dear reviewer findings of the demographic analysis is given in section
3.2. Sampling and data collection.
Reviewer Comments
- What happens if I want to add or remove an item from a construct?
8 Author Response: Dear reviewer we need to delete item only when loading is lower than 0.60 to achieve construct reliability and convergent validity of the construct. In manuscript constructs reliability is achieved following the values of (α) and composite reliability wherein CR and Cronbach's alpha values should be higher than .70 reflecting adequate construct reliability and validity.
Thanks.
Reviewer 5 Report
Dear authors,
The topic of your article is very interesting and the paper is well written. However, I suggest to improve it according to the following observations:
- In the introduction part, there are some paragraphs with no references in the text. For example: "In current situation wherein COVID-19 pandemic has left devastated impact global economy and badly affected supply chain operation, organizations are now seeking resilient kind of strategies to confront unforeseen events."This paragraph represent your own statement?
-
Product and marketing channel diversification could contribute to the economic success of small-scale companies involved in short food supply chains after the outbreak of the COVID-19 pandemic. I suggest to read the following article: Benedek, Z., Fertő, I., Galamba Marreiros, C., Aguiar, P. M. D., Pocol, C. B., Čechura, L., ... & Bakucs, Z. (2021). Farm diversification as a potential success factor for small-scale farmers constrained by COVID-related lockdown. Contributions from a survey conducted in four European countries during the first wave of COVID-19. PloS one, 16(5), e0251715.
-
The authors should use more recent references in the paper.
-
The objective of your study was to examine supply chain resilience and sustainable chain performance. Starting to this, you should define in the introduction part "the sustainable supply chain" and you should insert a paragraph in the literature review section about sustainability aspects.
-
Convenience sampling method has severe limitations. The authors should explain these limitations in the article.
-
Please mention the exact period of data collection. It is necessary to provide the questionnaire as supplementary file.
-
The results are well presented, but not enough discussed. There are a lot of published papers during the Covid 19 crisis related to resilience of supply chain. The authors should compare their own results with the results obtained by other scientists.
Best wishes,
The reviewer.
Author Response
Reviewer Comments to Author & Author response to Reviewer 5
Respected Reviewers
With reference to Manuscript “Investigating the Drivers of Supply Chain Resilience in the Wake of COVID-19 Pandemic: Empirical Evidence from Emerging Economy" author has reviewed article as per your suggestion.
Reviewer Comments
- In the introduction part, there are some paragraphs with no references in the text. For example: "In current situation wherein COVID-19 pandemic has left devastated impact global economy and badly affected supply chain operation, organizations are now seeking resilient kind of strategies to confront unforeseen events." This paragraph represents your own statement?
Author Response: Dear Reviewer statement is derived from
(Karmaker et al., 2021).
- Karmaker, C. L., Ahmed, T., Ahmed, S., Ali, S. M., Moktadir, M. A., & Kabir, G. (2021). Improving supply chain sustainability in the context of COVID-19 pandemic in an emerging economy: Exploring drivers using an integrated model. Sustainable production and consumption, 26, 411-427.
Reviewer Comments
- Product and marketing channel diversification could contribute to the economic success of small-scale companies involved in short food supply chains after the outbreak of the COVID-19 pandemic. I suggest to read the following article: Benedek, Z., Fertő, I., Galamba Marreiros, C., Aguiar, P. M. D., Pocol, C. B., Čechura, L., ... & Bakucs, Z. (2021). Farm diversification as a potential success factor for small-scale farmers constrained by COVID-related lockdown. Contributions from a survey conducted in four European countries during the first wave of COVID-19. PloS one, 16(5), e0251715.
2 Author comments: Author has done reading of suggested article and is citied inside the text.
Benedek, Z., Fertő, I., Galamba Marreiros, C., Aguiar, P. M. d., Pocol, C. B., Čechura, L., . . . Bakucs, Z. (2021). Farm diversification as a potential success factor for small-scale farmers constrained by COVID-related lockdown. Contributions from a survey conducted in four European countries during the first wave of COVID-19. PloS one, 16(5), e0251715.
Reviewer Comments
- The authors should use more recent references in the paper.
3 Author Comments: In the context of COVID-19 latest research work is as follows:
Benedek, Z., Fertő, I., Galamba Marreiros, C., Aguiar, P. M. d., Pocol, C. B., Čechura, L., . . . Bakucs, Z. (2021). Farm diversification as a potential success factor for small-scale farmers constrained by COVID-related lockdown. Contributions from a survey conducted in four European countries during the first wave of COVID-19. PloS one, 16(5), e0251715.
Karmaker, C. L., Ahmed, T., Ahmed, S., Ali, S. M., Moktadir, M. A., & Kabir, G. (2021). Improving supply chain sustainability in the context of COVID-19 pandemic in an emerging economy: Exploring drivers using an integrated model. Sustainable production and consumption, 26, 411-427.
Lopes de Sousa Jabbour, A. B., Chiappetta Jabbour, C. J., Hingley, M., Vilalta-Perdomo, E. L., Ramsden, G., & Twigg, D. (2020). Sustainability of supply chains in the wake of the coronavirus (COVID-19/SARS-CoV-2) pandemic: lessons and trends. Modern Supply Chain Research and Applications, 2(3), 117-122. doi: 10.1108/mscra-05-2020-0011
Rahi, S., Khan, M. M., & Alghizzawi, M. (2021). Factors influencing the adoption of telemedicine health services during COVID-19 pandemic crisis: an integrative research model. Enterprise Information Systems, 15(6), 769-793. doi: 10.1080/17517575.2020.1850872
Paul, S. K., Chowdhury, P., Moktadir, M. A., & Lau, K. H. (2021). Supply chain recovery challenges in the wake of COVID-19 pandemic. Journal of Business Research, 136, 316-329.
Belhadi, A., Kamble, S., Jabbour, C. J. C., Gunasekaran, A., Ndubisi, N. O., & Venkatesh, M. (2021). Manufacturing and service supply chain resilience to the COVID-19 outbreak: Lessons learned from the automobile and airline industries. Technological Forecasting and Social Change, 163, 120447.
Butt, A. S. (2021). Strategies to mitigate the impact of COVID-19 on supply chain disruptions: a multiple case analysis of buyers and distributors. The International Journal of Logistics Management, ahead-of-print(ahead-of-print). doi: 10.1108/ijlm-11-2020-0455
Reviewer Comments
- The objective of your study was to examine supply chain resilience and sustainable chain performance. Starting to this, you should define in the introduction part "the sustainable supply chain" and you should insert a paragraph in the literature review section about sustainability aspects.
- Author Comments: Dear reviewer in introduction both concepts are given and detail is as follows
If a firm face upheavals in supply chain operations and continue to perform that situation is characterize by resilience (Bhamra, Dani, & Burnard, 2011). Supply chain resilience is defined as operational capacity of a firm to return to its initial state after being disrupted and be stronger than before in a supply chain process (Bode, Wagner, Petersen, & Ellram, 2011). The importance of supply chain resilience is highlighted in earlier studies Blackhurst, Dunn, and Craighead (2011); Bode et al. (2011); Brandon‐Jones et al. (2014); Karmaker et al. (2021). According to Brandon‐Jones et al. (2014) firms are facing more disruption due to natural and manmade events and therefore resilience phenomenon should be investigated to understand how resilience help firms to recover quickly after disruption.
Therefore, current research fills the research gap and develops an integrated supply chain resilience model with the combination of factors such as supply chain intelligence, supply chain communication, leadership commitment, risk management orientation, supply chain capability, network complexity and big data analytics to investigate supply chain resilience during COVID 19 pandemic and sustainability supply chain performance in post pandemic context.
Reviewer Comments
- Convenience sampling method has severe limitations. The authors should explain these limitations in the article.
5 Author Comments: Although convenience sampling approach is supported by prior studies nevertheless following your suggestion author has updated it in limitation section.
Data were collected through convenience sampling approach which is non-probability sampling approach. Future researchers are suggested to collect data through probability sampling approach to mitigate any kind of sampling risk.
Reviewer Comments:
- Please mention the exact period of data collection. It is necessary to provide the questionnaire as supplementary file.
Author comments: Dear reviewer time period of data collection is updated in manuscript.
Reviewer Comments:
7.The results are well presented, but not enough discussed. There are a lot of published papers during the Covid 19 crisis related to resilience of supply chain. The authors should compare their own results with the results obtained by other scientists.
7 Author Response: Dear reviewer I do agree, the purpose of precise information was to maintain contents limit as per journal requirement, if I add further information word limit will exceed. Nevertheless, results are compared and contrast with previous studies in discussion section.
Belhadi, A., Kamble, S., Jabbour, C. J. C., Gunasekaran, A., Ndubisi, N. O., & Venkatesh, M. (2021). Manufacturing and service supply chain resilience to the COVID-19 outbreak: Lessons learned from the automobile and airline industries. Technological Forecasting and Social Change, 163, 120447.
Thanks.
Round 2
Reviewer 2 Report
I find this revision unsatisfactory as it leaves open most of the raised issues.
Author Response
Reviewer Comments to Author & Author response to Reviewer
Respected Reviewers/Editor
With reference to Manuscript “Investigating the Drivers of Supply Chain Resilience in the Wake of COVID-19 Pandemic: Empirical Evidence from Emerging Economy", author has answered reviewer comments in following section.
Reviewer 2 Comments to Author and Author response to reviewer
Reviewer Comments:
1 Average variance extracted (AVE) is not reliable for structural equation modelling. See Henseler, J., Ringle, C. M., Sarstedt, M., 2014. A new criterion for assessing discriminant validity in variance-based structural equation modeling. Journal of the Academy of Marketing Science 43 (1), 115–135) Therefore the author should prove his case.
Author Response:
Dear reviewer, I would be thankful if I get further guidance in which part of structural equation modeling the role of AVE is not reliable?
Is it in measurement model? If yes then within measurement model AVE has been used to assess convergent validity of the constructs. And similarly, AVE has used in discriminant validity.
For discriminant validity square root of AVE is used to measure discriminant validity exhibited in Table 3.
Concerning with new method provided by Henseler, J., Ringle, C. M., & Sarstedt, M. (2015), author has done reading and assessed discriminant validity through Heterotrait monotrait (HTMT) analysis.
Henseler, J., Ringle, C. M., & Sarstedt, M. (2015). A new criterion for assessing discriminant validity in variance-based structural equation modeling. Journal of the Academy of Marketing Science, 43(1), 115-135.
The results of the HTMT analysis are shown in Table 5.
Reviewer Comments:
- PLS can be applied as an alternative to multiple linear regression (MLR) because the latter fails “when the factors are few in number, are not significantly redundant (collinear), and have a well-understood relationship to the responses, then can be a good way to turn data into information)“ Therefore the authors should show that this is the case Tobias, R. D. (1995, April). An introduction to partial least squares regression. In Proceedings of the twentieth annual SAS users group
Author Response:
Dear Reviewer, the partial least square approach is applied in line with F. Hair Jr, J., Sarstedt, M., Hopkins, L., &G. Kuppelwieser, V. (2014) suggested that PLS approach is appropriate to be taken into data analysis wherein the objective of research is towards theory developing instead of theory testing.
- Hair Jr, J., Sarstedt, M., Hopkins, L., & G. Kuppelwieser, V. (2014). Partial least squares structural equation modeling (PLS-SEM) An emerging tool in business research. European Business Review, 26(2), 106-121.
Further support is taken from Rahi, S. (2017). Structural Equation Modeling Using SmartPLS. CreateSpace Independent Publishing Platform
Author Rahi, S. (2017) stated that PLS approach appropriate in less matured model. As this research model is an integrated model that combines several factors hence, author has selected VB-SEM to conduct data analysis.
Reviewer Comments:
- Regarding effect size the author may refer to Cohen's term d and show the result. See Cohen, Jacob. "Things I have learned (so far)." In Annual Convention of the American Psychological Association, 98th, Aug, 1990, Boston, MA, US; Presented at the aforementioned conference.
American Psychological Association, 1992. Ellis, P. D. (2010). The essential guide to effect sizes: Statistical power, meta-analysis, and theinterpretation of research results. Cambridge university press
Author Response:
Dear reviewer for effect size analysis author acknowledged your point and suggested reference is added.
Cohen, J. (1992). Things I have learned (so far).
Annual Convention of the American Psychological Association, 98th, Aug, 1990, Boston, MA, US; Presented at the aforementioned conference.
Reviewer Comments:
- Regarding the case in which the regressand is not a continuous variable but a state, which may or may not hold, or a category in a given classification the author should mention why he does not use logit analysis. See Cramer, J. S. (2003). Logit models from economics and other fields. Cambridge University Press. Orlando, G., & Pelosi, R. (2020). Non-Performing Loans for Italian Companies: When Time Matters. An Empirical Research on Estimating Probability to Default and Loss Given Default. International Journal of Financial Studies, 8(4), 68.
Author Response:
Respected reviewer logit model is not applicable in this research. This research is kind of behavioral and test causal relationship among variables. Overall positivist paradigm is followed to design research model as suggested by Rahi, S. (2017).
Rahi, S. (2017). Research design and methods: A systematic review of research paradigms, sampling issues and instruments development. International Journal of Economics & Management Sciences, 6(2), 1-5.
In addition to that researcher has used some additional test to see predictive power of the research model such as blindfolding analysis using consistent with
Geisser, S. (1974). A predictive approach to the random effect model. Biometrika, 61(1), 101-107.
Reviewer Comments:
5 The layout is wrong, for example figures and tables are bigger
than text width
Author Response:
Copyediting of this manuscript is yet to be done. Problem occurs due to template and author has requested to editor for assistance. In final version, this issue will be resolved.
Reviewer Comments:
- Few spelling mistakes, for example “commination channels” perhaps should be “communication channels”
Author Response:
Author has resolved this issue and highlighted in text body spellings are fixed.
Reviewer Comments:
- The paper seems to reference disproportionately some authors that are not very close to get a Nobel prize, for example Rahi is cited 10 times. The author should review carefully the bibliography.
Author Response: The only objective to cite highlighted author is to improve methodology section with latest research. Author found useful methods from Rahi et al and hence citied multiple times. Following are few references which supports to research methodology and therefore citied in this research.
Rahi, S., Ghani, M., & Ngah, A. (2018). A structural equation model for evaluating user’s intention to adopt internet banking and intention to recommend technology. Accounting, 4(4), 139-152.
Rahi, S. (2017). Research design and methods: A systematic review of research paradigms, sampling issues and instruments development. International Journal of Economics & Management Sciences, 6(2), 1-5.
Rahi, S. (2017). Structural Equation Modeling Using SmartPLS. CreateSpace Independent Publishing Platform. https://books.google.com.my/books?id = XwF6tgEACAAJ
Hair, J. F., Ringle, C. M., & Sarstedt, M. (2013). Partial least squares structural equation modeling: Rigorous applications, better results and higher acceptance.
Hair Jr, J. F., Hult, G. T. M., Ringle, C., & Sarstedt, M. (2016). A primer on partial least squares structural equation modeling (PLS-SEM). Sage Publications.
Thanks.

Reviewer 4 Report
Dear Authors,
Thanks for your efforts. The quality has been enhanced.
Now my main point is one serious objection: 10/11 articles have been cited from a single author (Rahi, S: 34-43). It is unacceptable.
Author Response
Reviewer Comments to Author & Author response to Reviewer 4
Respected Reviewers
With reference to Manuscript “Investigating the Drivers of Supply Chain Resilience in the Wake of COVID-19 Pandemic: Empirical Evidence from Emerging Economy" author has reviewed article as per your suggestion.
Reviewer Comments
1: Comments and Suggestions for Authors
Dear Authors,
Thanks for your efforts. The quality has been enhanced.
Now my main point is one serious objection: 10/11 articles have been cited from a single author (Rahi, S: 34-43). It is unacceptable
1 Author Response: Dear reviewer as per your recommendation author citations issue is resolved. Concerning with author Rahi et now citations are reduced to 6 instead of 11.
Thanks.
Round 3
Reviewer 2 Report
I find this revision unsatisfactory as it leaves open most of the raised issues.
Author Response
Reviewer Comments to Author & Author response to Reviewer
With reference to Manuscript “Investigating the Drivers of Supply Chain Resilience in the Wake of COVID-19 Pandemic: Empirical Evidence from Emerging Economy", author comments are as follows:
Comments from Reviewer 2
1 Average variance extracted (AVE) is not reliable for structural equation modelling. See Henseler, J., Ringle, C. M., Sarstedt, M., 2014. A new criterion for assessing discriminant validity in variance-based structural equation modeling. Journal of the Academy of Marketing Science 43 (1), 115–135) Therefore the author should prove his case.
Author Response:
Dear reviewer, I would be thankful if I get further guidance in which part of structural equation modeling the role of AVE is not reliable?
Is it in measurement model? If yes then within measurement model AVE has been used to assess convergent validity of the constructs. And similarly, AVE has used in discriminant validity.
For discriminant validity square root of AVE is used to measure discriminant validity exhibited in Table 3.
Concerning with new method provided by Henseler, J., Ringle, C. M., & Sarstedt, M. (2015), author has done reading and assessed discriminant validity through Heterotrait monotrait (HTMT) analysis.
Henseler, J., Ringle, C. M., & Sarstedt, M. (2015). A new criterion for assessing discriminant validity in variance-based structural equation modeling. Journal of the Academy of Marketing Science, 43(1), 115-135.
The results of the HTMT analysis are shown in Table 5.
Reviewer Comments:
- PLS can be applied as an alternative to multiple linear regression (MLR) because the latter fails “when the factors are few in number, are not significantly redundant (collinear), and have a well-understood relationship to the responses, then can be a good way to turn data into information)“ Therefore the authors should show that this is the case Tobias, R. D. (1995, April). An introduction to partial least squares regression. In Proceedings of the twentieth annual SAS users group
Author Response:
Dear Reviewer, the partial least square approach is applied in line with F. Hair Jr, J., Sarstedt, M., Hopkins, L., &G. Kuppelwieser, V. (2014) suggested that PLS approach is appropriate to be taken into data analysis wherein the objective of research is towards theory developing instead of theory testing.
- Hair Jr, J., Sarstedt, M., Hopkins, L., & G. Kuppelwieser, V. (2014). Partial least squares structural equation modeling (PLS-SEM) An emerging tool in business research. European Business Review, 26(2), 106-121.
Further support is taken from Rahi, S. (2017). Structural Equation Modeling Using SmartPLS. CreateSpace Independent Publishing Platform
Author Rahi, S. (2017) stated that PLS approach appropriate in less matured model. As this research model is an integrated model that combines several factors hence, author has selected VB-SEM to conduct data analysis.
Reviewer Comments:
- Regarding effect size the author may refer to Cohen's term d and show the result. See Cohen, Jacob. "Things I have learned (so far)." In Annual Convention of the American Psychological Association, 98th, Aug, 1990, Boston, MA, US; Presented at the aforementioned conference.
American Psychological Association, 1992. Ellis, P. D. (2010). The essential guide to effect sizes: Statistical power, meta-analysis, and theinterpretation of research results. Cambridge university press
Author Response:
Dear reviewer for effect size analysis author acknowledged your point and suggested reference is added.
Cohen, J. (1992). Things I have learned (so far).
Annual Convention of the American Psychological Association, 98th, Aug, 1990, Boston, MA, US; Presented at the aforementioned conference.
Reviewer Comments:
- Regarding the case in which the regressand is not a continuous variable but a state, which may or may not hold, or a category in a given classification the author should mention why he does not use logit analysis. See Cramer, J. S. (2003). Logit models from economics and other fields. Cambridge University Press. Orlando, G., & Pelosi, R. (2020). Non-Performing Loans for Italian Companies: When Time Matters. An Empirical Research on Estimating Probability to Default and Loss Given Default. International Journal of Financial Studies, 8(4), 68.
Author Response:
Respected reviewer logit model is not applicable in this research. This research is kind of behavioral and test causal relationship among variables. Overall positivist paradigm is followed to design research model as suggested by Rahi, S. (2017).
Rahi, S. (2017). Research design and methods: A systematic review of research paradigms, sampling issues and instruments development. International Journal of Economics & Management Sciences, 6(2), 1-5.
In addition to that researcher has used some additional test to see predictive power of the research model such as blindfolding analysis using consistent with
Geisser, S. (1974). A predictive approach to the random effect model. Biometrika, 61(1), 101-107.
Reviewer Comments:
5 The layout is wrong, for example figures and tables are bigger
than text width
Author Response:
Copyediting of this manuscript is yet to be done. Problem occurs due to template and author has requested to editor for assistance. In final version, this issue will be resolved.
Reviewer Comments:
- Few spelling mistakes, for example “commination channels” perhaps should be “communication channels”
Author Response:
Author has resolved this issue and highlighted in text body spellings are fixed.
Reviewer Comments:
- The paper seems to reference disproportionately some authors that are not very close to get a Nobel prize, for example Rahi is cited 10 times. The author should review carefully the bibliography.
Author Response: The only objective to cite highlighted author is to improve methodology section with latest research. Author found useful methods from Rahi et al and hence citied multiple times. Following are few references which supports to research methodology and therefore citied in this research.
Rahi, S., Ghani, M., & Ngah, A. (2018). A structural equation model for evaluating user’s intention to adopt internet banking and intention to recommend technology. Accounting, 4(4), 139-152.
Rahi, S. (2017). Research design and methods: A systematic review of research paradigms, sampling issues and instruments development. International Journal of Economics & Management Sciences, 6(2), 1-5.
Rahi, S. (2017). Structural Equation Modeling Using SmartPLS. CreateSpace Independent Publishing Platform. https://books.google.com.my/books?id = XwF6tgEACAAJ
Hair, J. F., Ringle, C. M., & Sarstedt, M. (2013). Partial least squares structural equation modeling: Rigorous applications, better results and higher acceptance.
Hair Jr, J. F., Hult, G. T. M., Ringle, C., & Sarstedt, M. (2016). A primer on partial least squares structural equation modeling (PLS-SEM). Sage Publications.
Thanks.
Reviewer 4 Report
The paper can be accepted.